

# Article processing charges for open access publication—the situation for research intensive universities in the USA and Canada

David Solomon[1] and Bo-Christer Björk[2]

[1] Internal Medicine/Office of Medical Education Research and Development, Michigan State University, E Lansing, MI, United States
[2] Information Systems Science, Hanken School of Economics, Helsinki, Finland

## ABSTRACT

**Background.** Open access (OA) publishing via article processing charges (APCs) is growing as an alternative to subscription publishing. The Pay It Forward (PIF) Project is exploring the feasibility of transitioning from paying subscriptions to funding APCs for faculty at research intensive universities. Estimating of the cost of APCs for the journals authors at research intensive universities tend to publish is essential for the PIF project and similar initiatives. This paper presents our research into this question.

**Methods.** We identified APC prices for publications by authors at the 4 research intensive United States (US) and Canadian universities involved in the study. We also obtained APC payment records from several Western European universities and funding agencies. Both data sets were merged with Web of Science (WoS) metadata. We calculated the average APCs for articles and proceedings in 13 discipline categories published by researchers at research intensive universities. We also identified 41 journals published by traditionally subscription publishers which have recently converted to APC funded OA and recorded the APCs they charge.

**Results.** We identified 7,629 payment records from the 4 European APC payment databases and 14,356 OA articles authored by PIF partner university faculty for which we had listed APC prices. APCs for full OA journals published by PIF authors averaged 1,775 USD; full OA journal APCs paid by Western European funders averaged 1,865 USD; hybrid APCs paid by Western European funders averaged 2,887 USD. The APC for converted journals published by major subscription publishers averaged 1,825 USD. APC funded OA is concentrated in the life and basic sciences. APCs funded articles in the social sciences and humanities are often multidisciplinary and published in journals such as PLOS ONE that largely publish in the life sciences.

**Conclusions.** Full OA journal APCs average a little under 2,000 USD while hybrid articles average about 3,000 USD for publications by researchers at research intensive universities. There is a lack of information on discipline differences in APCs due to the concentration of APC funded publications in a few fields and the multidisciplinary nature of research.

Corresponding author
David Solomon,
dsolomon@msu.edu

## INTRODUCTION

Since the launch of the first Open Access (OA) journals funded by Article Processing Charges (APC) around 2000, APC funded OA publication has grown rapidly. By 2010 the number of articles published in APC funded OA journals indexed in Scopus surpassed the number of articles published in OA journals funded by other means (*Solomon, Laakso & Björk, 2013*). There also is evidence that APC funded OA articles are continuing to grow exponentially. Between 2010 and 2012, the number of APC funded OA articles published by 7 major OA publishers more than doubled from 41,974 to 87,021 (*Neylon, 2013*). Along with publishers that only publish APC funded OA journals, large, traditionally subscription publishers, are rapidly increasing the number of OA journals they publish. For example, between August 2013 and June 2016 Elsevier increased the number of APC funded OA journals they publish from 46 to over 550[1] (*Solomon & Björk, 2012*).

As publishing in APC funded OA articles becomes more commonplace there is concern that if libraries begin paying publishing fees in lieu of subscriptions it could become a significant burden for libraries at research intensive universities. A recent survey of libraries found about 20% of the funding for APCs is coming out of library budgets with 70% of the respondents indicating the funding for APCs at their libraries is coming out of the materials budget (*Lara, 2014*). There is a real concern such a significant shift in funding for scholarly publishing would be unsustainable for research intensive universities in the USA and Canada. The University of California (UC), Davis, is leading a multi-institutional project titled Pay It Forward (PIF) including 4 research intensive universities focused on estimating the likely budgetary impact of such a transition. The research is being funded by the Andrew W. Mellon Foundation (*Smith, 2014*). As consultants on this project, our major role was in helping estimate the likely cost of APCs for articles published by researchers at research intensive universities in the USA and Canada. This paper summarizes our findings in this area.

There have been several studies that have attempted to characterize the cost of APC funded OA. In 2012 we conducted a comprehensive review of the APC prices for the journals in the Directory of Open Access Journals (DOAJ) that were listed in the directory as charging APCs (*Solomon & Björk, 2012*). As part of the study we collected the article counts for 2011, the most recent calendar year. We gathered either the listed APC price off the journal web site or our best estimate of the typical APC price when there was not a specific single APC listed. We found that across this broad range of journals the APC prices both raw and weighted by the number of articles published to be around 900 USD.

In 2014 we revisited a subset of the journals included in our previous study. We attempted to limit the sample to those journals which researchers at research intensive universities in the US, Canada and Western Europe would likely publish by selecting only journals from publishers with at least 8 journals of which at least 2 were indexed in the (WoS). We included all APC funded journals from publishers who met the criteria above with less than 30 journals and randomly sampled 30 journals from publishers which published over 30 journals. This resulted in a sample of 187 journals from 9 publishers. We found the number of articles published in these journals increased between 2011 and 2013 by an

[1] Based on fully OA journals listed by Elsevier at https://www.elsevier.com/about/open-science/open-access/open-access-journals on 2016-06-20.

average 24.5% even when PLOS ONE was left out of the analysis as an outlier. The average APC in this subset of journals was 1,292 USD in the fall of 2012 and had increased to 1,418 USD by the fall of 2014.

Most of the large, traditionally subscription publishers have begun publishing full OA journals We collected APC prices for 102 journals published by 6 major traditionally subscription publishers (*Björk & Solomon, 2014*). The 102 journals published by major traditionally subscription publishers were on average 679 USD higher than our sample of journals from full OA publishers. Interestingly 15 of the 102 journals from these major publishers had APCs under 500 USD. Many of these journals however were medical journals that only published case reports (*Cohen, 2006*).

Morrison and colleagues (*2015*) conducted a study gathering the list price and pricing methodology of the journals in the DOAJ that charge APCs. They used a stratified sampling procedure that selected 1,584 of the 2,567 journals listed in the DOAJ as charging APCs. Their results were similar to our first study finding an average APC of 964 USD suggesting there has been modest inflation in APC prices in the 3 years in between the 2 studies. As with our first study, this study included a wide range of journals many of which authors from research intensive universities in the US, Canada and Western Europe are unlikely to publish.

The previous studies focused on the published prices of full OA journals. Pinfield and his colleagues (*2016*) conducted a study assessing the total cost to institutions of paying both subscriptions and APCs including APCs from hybrid journals which are subscription journals where authors can pay APCs to make their individual articles OA. They used data from 23 universities in the United Kingdom (UK) gathered between 2007 and the first quarter of 2014. Pinfield and his colleagues also attempted to estimate the administrative costs of paying APCs. They found a significant increase in the total costs to these universities following policy changes in the UK encouraging APC funded OA. By 2013, the APCs paid by these universities to major subscription publishers for hybrid articles increased the total cost of access to these journals by about 10%. They defined the total cost of access as APCs paid for their authors, subscription fees and the administrative costs of paying APCs. They also found it difficult to estimate administration costs and these costs appeared to vary considerably among universities. The APC levels Pinfield and his colleagues found were roughly consistent with our earlier study.

While the studies described above begin to provide a picture of APC pricing and in the case of Pinfield and his colleagues' study, the total cost of access for of UK universities, we felt we needed additional data that would be more directly applicable for estimating the feasibility of transitioning to APC funded OA for research intensive institutions in the USA and Canada. We believe there is evidence that researchers at research intensive universities tend to publish in the more expensive APC journals and hence previous studies estimating APC prices that included all OA journal would underestimate the cost of APCs for this group of researchers.

The goal of the study was to estimate the per article APC expenditures for the publications of researchers at research intensive universities in the USA and Canada.

## METHODS

We used three types of data to characterize what the likely cost of APCs would be for research intensive universities in the USA and Canada. Each has its strengths and limitations. By triangulating different sources of information we felt we could derive a more robust estimate of the likely cost of APCs for researchers at these institutions. Firstly, we tied published APC prices to OA articles published by faculty at the PIF partner universities. Secondly we identified a sample of journals published by major traditionally subscription publishers that were recently converted from subscription to APC funded OA. Thirdly we gathered APC payments made by funding agencies and universities from special budgets set aside for this purpose such as was used by Pinfield and his colleagues in their study of UK university APC funding programs.

**WoS Metadata**—Thomson Reuters in partnership with the PIF Project provided article level metadata from the WoS for the articles published by the PIF partner universities between 2009 and 2013. They also provided article level metadata for the APC payment records we obtained from the universities and funding agencies described below that were matched via Digital Object Identifiers (DOIs). The metadata contained a variety of useful information however for the purposes this analysis we focused on the type of publication limiting the analysis to research articles and conference proceedings. We also broke down the payment and APC pricing results by discipline. The PIF project settled on a 13 category discipline coding scheme derived from Thomson Reuters Essential Science Indicators (ESI) and Scopus's 23 category discipline scheme. Since there is no ESI code for the arts and humanities, we coded articles and proceedings listed in the Arts & Humanities Citation Index (AHCI) as being in the arts and humanities. The coding scheme developed by the project is shown in Table 3.

**APC Prices for Publications by PIF Partner University Faculty**—As noted earlier, to get an accurate picture of the cost of transitioning from subscription to APC funded OA, we needed to characterize the APC prices for the types of OA journals researchers at research intensive universities in the USA and Canada would likely publish. We matched the articles and proceedings researchers at the 4 PIF partner institutions published between 2009 and 2013 obtained from the WoS with APC pricing data obtained from journal websites collected in 2014 by Morrison and her colleagues. Their pricing data was the most up-to-date and comprehensive data we were able to locate and they had made it available in a public archive. We used International Standard Serial Numbers (ISSN) to match the individual articles or proceedings identified in the WoS with the APC prices for the journals in which they were published.

**Subscription journals converting to APCs**—We attempted to identify journals from major publishers that traditionally published subscription journals which had transitioned from subscription to APC funded OA. To identify these journals, we searched the websites of 7 large publishers as well as the Internet for press releases, blog entries and other indications journals from these publishers flipped from subscription to OA. Once we identified that a journal had "flipped" to OA, we gathered the APC and other metadata from the journal web site.

**APC payment repositories**—We were able to obtain APC payment data from 4 sources. All were based in Europe. These included UK universities; German universities and the Max Plank Digital Library; the Austrian Science Fund (FWF) and the Wellcome Trust. The data were downloaded around March 1, 2015. The specific information provided, requirements, rules for payment, time period in which the payments were made and the currency differed among these data sets. Each is described in more detail below:

*United Kingdom (UK) Universities*—Stuart Lawson and his colleagues at Jisc compiled APC payment data from a number of UK universities (*House of Commons, 2014*; *Lawson, 2014*). We combined the two overlapping data sets removing the duplicates. The payments were converted from GBP to USD using an exchange rate of 1.6 which roughly reflected the exchange rate during the period the APC payments were made. The data include both full OA and hybrid payments.

*Wellcome Trust*—The Wellcome Trust maintains a special budget for paying publication charges for the research it funds. The Trust has released APC payments made during their 2012–2013 and 2013–2014 fiscal years (*Kiley, 2014*; *Kiley, 2015*). As with the UK university data above, a currency conversion rate of 1.6 was used for converting from GBP to USD. The data include both full OA and hybrid payments.

*German Universities and Foundations*—APC payment records were available for 22 German universities and 5 other participating institutions (*Apel et al., 2015*). The payment data in EUR were converted to USD using an exchange rate of 1.3 which roughly reflected the exchange rate during the period the payments were made. Payments were only made for publication in fully OA journals.

*Austrian Science Fund (FWF)*—The FWF covers the cost of APCs and other publication charges for researchers they fund. The data for 2013 was available at the time we merged the data with WoS (*Reckling & Kenzian, 2014*). Unfortunately 2014 data became available just after we requested WoS metadata from Thomson Rueters (*Reckling & Rieck, 2015*). The data include both full OA and hybrid payments,

The data from these universities and funding agencies was merged with WoS metadata in late April 2015 using DOIs.

## RESULTS

We identified 14,356 OA articles and proceeding published by researchers at PIF partner universities between 2009 and 2013 in OA journals that we were able to obtain APC prices. Please note the article/proceedings were published between 2009 and 2013 while the APC prices were gathered in 2014.

We collected a total of 13,819 payment records from the 4 APC payment databases. A total of 12,172 or 88% were matched with WoS metadata based on DOIs. After removing duplicates, records that were not articles or proceedings or were missing key information, there were 7,629 payment records that were used in the analyses described below.

Table 1 below presents hybrid and full OA payments from the European payment databases and full APC prices for articles and proceedings by authors from PIF universities. The results are broken down by the 13 discipline categories.

**Table 1  Breakdown of different sources of APC payment/charges by discipline.** APCs in US dollars.

| Discipline | Hybrid payments | | | Full OA payments | | | Full OA prices | | |
|---|---|---|---|---|---|---|---|---|---|
| | Payments | *N* | SD | Payments | *N* | SD | Charges | *N* | SD |
| Arts and humanities | 2,168.26 | 5 | 1,276.86 | *No data* | 0 | 0 | 1,273.26 | 19 | 354.76 |
| Multidisciplinary | 2,074.42 | 16 | 1,631.24 | 1,896.48 | 64 | 1,355.18 | 1,345.83 | 522 | 50.39 |
| Mathematics | 2,579.93 | 52 | 908.46 | 905.60 | 5 | 455.97 | 1,209.79 | 24 | 69.60 |
| Clinical Medicine | 3,000.33 | 626 | 1,082.86 | 1,870.32 | 526 | 584.89 | 1,753.60 | 3,456 | 466.20 |
| Biomedical Research | 2,996.56 | 1,377 | 1,212.28 | 1,952.02 | 1,076 | 864.70 | 1,830.36 | 5,511 | 552.38 |
| Life Sciences | 2,859.62 | 667 | 1,164.24 | 1,876.85 | 579 | 716.09 | 1,789.30 | 2,286 | 552.35 |
| Chemistry | 2,901.43 | 370 | 915.12 | 2,403.16 | 47 | 1,629.68 | 1,712.00 | 189 | 308.93 |
| Physics and Astronomy | 2,575.06 | 241 | 844.62 | 1,890.44 | 190 | 1,395.89 | 1,327.90 | 139 | 84.72 |
| Engineering | 2,718.00 | 365 | 903.61 | 1,669.40 | 97 | 737.46 | 1,900.44 | 436 | 453.47 |
| Earth Science | 2,905.81 | 264 | 824.92 | 1,523.47 | 164 | 706.69 | 1,599.72 | 664 | 331.82 |
| Business and Economics | 2,521.58 | 35 | 931.65 | 1,415.65 | 4 | 101.74 | 1,350.00 | 11 | 0.00 |
| Psychiatry/Psychology | 2,955.87 | 204 | 956.31 | 1,647.01 | 231 | 582.40 | 1,787.35 | 373 | 433.94 |
| Social Science | 2,736.35 | 307 | 878.52 | 1,822.51 | 117 | 407.03 | 1,940.57 | 726 | 460.28 |
| Total | 2,886.88 | 4,529 | 1,076.15 | 1,864.53 | 3,100 | 838.55 | 1,775.07 | 14,356 | 510.65 |

The results presented in Table 1 by discipline should be interpreted with caution. There were very few publications in some of disciplines including arts and humanities, mathematics and business/economics. In addition, many of the articles/proceedings coded in disciplines such as the arts, humanities and social sciences are in journals that would generally be considered to be in the biomedical or life sciences. Table 4 presents the number and percentage of articles/proceedings in each journal within each discipline. For example, as can be seen in Table 4, 59% of the articles/proceedings in engineering were published in BMC Bioinformatics and 68% of the article/proceedings in the arts and humanities where published in PLOS ONE. We believe in most cases these article/proceedings describe research that is multidisciplinary but the ESI/AHCI coding scheme we used only assigned a single discipline code to each publication record. For example, one of the articles coded as being in the arts and humanities was titled "Effects of Culture on Musical Pitch Perception" and was published in PLOS ONE.

We attempted to identify journals from large traditionally subscription publishing houses that have transitioned from subscription to APC funded OA. We were able to locate 41 such journals from 7 major publishers. A summary of the results is presented in Table 2.

With the exception of one outlier, Nature Communications, the APCs charged for these journals appear similar to APCs for APC funded OA journals published by fully OA publishers. Five of these journals, 2 each from Springer and Elsevier and 1 from Oxford University Press were part of the Sponsoring Consortium for Open Access Publishing in Particle Physics (SCOAP3) tendering process (*SCOAP³, 2015*). The APCs for these journals averaged 1,674 USD.
Table 2 **Breakdown of different sources of APC payment/charges by discipline.** APCs in US Dollars.

| Publisher | Mean | N | SD |
|---|---|---|---|
| De Gruyter | 1,356.00 | 5 | 309.46 |
| Elsevier | 1,950.00 | 7 | 485.63 |
| Nature Publishing Group | 5,200.00 | 1 | |
| Oxford University Press | 2,163.33 | 3 | 625.81 |
| Springer | 1,380.46 | 13 | 372.11 |
| Tailor & Francis | 1,031.67 | 3 | 451.12 |
| Wiley | 2,408.00 | 9 | 550.63 |
| Total | 1,825.20 | 41 | 829.68 |

**Notes.**
APCs reported in US dollars.

## DISCUSSION

We encountered several challenges in estimating APC prices for journals researchers at research intensive universities in the US and Canada are likely to publish.

- Assigning a single discipline category to each article/proceeding is somewhat artificial given research is often multidisciplinary. While there were two other discipline coding schemes at different levels of specificity in the WoS that assigned multiple discipline codes for each publication, attempting to use a coding scheme with multiple codes per publication to sort out disciplinary differences in APC prices would have been extremely complex and probably not helpful. The current APC market is also concentrated in a few disciplines and there are very few APC funded OA journals that publish in the social sciences and humanities. The articles and proceedings that we found coded in these disciplines were often multidisciplinary and published in journals that generally publish material in the life, medical or biological sciences.
- The APC market is complex and getting even more complex with full OA and hybrid APCs as well as an increasing number of comprehensive "cost of ownership" agreements negotiated between publishers and universities, university consortia and research funders. APC s are often complex with various discounts and fee structures for different types of publications (*Björk & Solomon , 2012*).
- There is very little information available on the costs associated with paying APCs, both for the publishers and the organizations that are funding APCs nor how these compare with the costs of negotiating and paying subscription fees.
- The current APC market is fluid and subject to market forces brought about by the APC payment policies of the universities, consortia and funding agencies that are increasingly paying APCs as this market evolves (*Björk, 2016*).

Despite these difficulties we found a pattern in APC list prices and APC payments by universities and funding agencies that were fairly consistent in all 3 sources of data we used in the study.

(1) For researchers at research intensive universities, APCs the paid for the fully OA journals average around 1,800 USD while hybrid journal APCs average about 3,000 USD.

(2) There do not appear to be large discipline differences in APCs. This likely reflects the limitations of the data available. There were very few publications in a number of disciples and those tended to be multidisciplinary.

(3) Based on the very small sample of journals flipped from subscription to OA by major traditionally subscription publishers that publish a major portion of the scholarly literature, the APCs for these journals appear to be similar to journals that were launched as full OA journals.

Our estimates of the APCs for fully OA journals are considerably higher than estimates of APCs in previous research. Our own and Morrison and her colleagues' estimates of APC charges (*Solomon & Björk, 2012*; *Morrison et al., 2015*) reflect the full distribution of OA journals in the DOAJ. Many of those journals with very low APCs are regional journals that researchers at research intensive universities in the USA, Canada and Western Europe are unlikely to publish. In our later study (*Björk & Solomon, 2014*) we did try to limit the journals included to those which researchers at research intensive universities would likely publish. However, our methodology for achieving this was weak, limiting the sample to journals published by OA publishers with at least 2 journals in the Web of Science. The criteria did result in a significantly higher APC estimate but well below estimates from this study. We feel this is largely due to the methodology used. Our current study used 3 separate approaches. One, probably the most robust, used APC prices for articles and proceedings authored by researchers at the 4 PIF partner universities. The second approach used actual APC payments made by 2 European foundations and the universities in 2 European countries for their researchers' publications. The third were APC prices for journals "flipped" from subscription to an APC business model by major traditionally subscription publishers. The APC estimates from these three methodologies triangulated at roughly 1,800 USD for articles published in full OA journals. We feel the estimates are probably the best available for APCs that would likely be paid currently for researchers at research intensive universities in the USA, Canada as well as Western Europe.

Many European governments and funding agencies are working towards transitioning all their research publications to OA with a preference for APC funded OA (*Schimmer, Geschuhn & Vogler, 2015*). The PIF project is modeling a similar transition for research intensive universities in the USA and Canada. At this juncture it appears we are moving towards a wide scale transition of the existing subscription journals to OA publishing much of it funded by APCs (*Shearer, 2016*). Having reasonable estimates of the likely costs of APCs is essential for modeling the cost of this large scale transition to OA scholarly publishing. The results presented in this paper are based on a number of sources of information and we feel despite their limitations, reflect the best data available for characterizing the per publication APC costs for research intensive universities in the USA, Canada and Western Europe.

# APPENDIX 1

**Table 3** Scopus—WOS subject mapping.

| Scopus 27 | Subject merge | ESI 23 |
|---|---|---|
| General | Multidisciplinary | E Multidisciplinary (sciences) |
| S Mathematics | Mathematics | E Mathematics |
| S Medicine | Clinical Medicine | E Clinical Medicine |
| S Pharmacology, toxicology and pharmaceutics | | E Pharmacology & Toxicology |
| S Nursing | | |
| S Health professions | | |
| S Dentistry | | |
| S Immunology and microbiology | Biomedical Research Disciplines | E Immunology |
| | | E Microbiology |
| S Biochemistry, genetics and molecular biology | | E Molecular Biology and Genetics |
| S Neuroscience | | E Neuroscience and Behavior |
| S Agricultural and biological sciences | Life Sciences | E Agricultural sciences |
| | | E Biology and Biochemistry |
| S Veterinary | | E Plant and animal sciences |
| S Chemistry | Chemistry | E Chemistry |
| S Chemical engineering | | |
| S Physics and astronomy | Physics and Astronomy | E Physics |
| | | E Space sciences |
| S Engineering | Engineering | E Engineering |
| S Materials Science | | E Materials Science |
| S Computer Science | | E Computer Science |
| S Energy | | |
| S Earth and planetary sciences | Earth Sciences | E Geosciences |
| S Environmental science | | E Environment/ecology |
| S Business management and accounting | Business and economics | E Economics and business |
| S Decisions sciences | | |
| S Economics, econometrics and Finance | | |
| S Psychology | Psychiatry/Psychology | E Psychiatry/Psychology |
| S Social Sciences | Social Science | E Social sciences, general |
| S Arts & Humanities | Arts and humanities | E (Arts and Humanities—category to be created from WOS categories/research areas. PIF team will have to assign journals with both an A &H and SocSci ESI category to a single preferred category) |

*

# APPENDIX 2

**Table 4** **Number of APC funded publications in each journal within each discipline based on PIF partner authored article/ proceeding 2009– 2013.** Article Processing Charge (APC) based on *Morrison et al. (2015)*.

| Discipline | Journal | APC | Number | Percent |
|---|---|---|---|---|
| **Arts and Humanities** | ENTROPY | 1,349 | 2 | 10.5% |
| | ENVIRONMENTAL RESEARCH LETTERS | 1,920 | 1 | 5.3% |
| | PLOS ONE | 1,350 | 13 | 68.4% |
| | RELIGIONS | 337 | 2 | 10.5% |
| | SCIENTIFIC REPORTS | 1,350 | 1 | 5.3% |
| **Multidisciplinary** | DISCRETE DYNAMICS IN NATURE AND SOCIETY | 1,200 | 1 | 0.2% |
| | PLOS ONE | 1,350 | 477 | 91.4% |
| | SCIENTIFIC REPORTS | 1,350 | 39 | 7.5% |
| | SCIENTIFIC WORLD JOURNAL | 1,200 | 1 | 0.2% |
| | SYMMETRY-BASEL | 562 | 2 | 0.4% |
| | THESCIENTIFICWORLDJOURNAL | 1,200 | 2 | 0.4% |
| **Mathematics** | ABSTRACT AND APPLIED ANALYSIS | 1,200 | 2 | 8.3% |
| | COMPUTATIONAL AND MATHEMATICAL METHODS IN MEDICINE | 1,200 | 16 | 66.7% |
| | FIXED POINT THEORY AND APPLICATIONS | 985 | 1 | 4.2% |
| | JOURNAL OF APPLIED MATHEMATICS | 1,200 | 2 | 8.3% |
| | PLOS ONE | 1,350 | 3 | 12.5% |
| **Clinical Medicine** | AFRICAN JOURNAL OF PHARMACY AND PHARMACOLOGY | 600 | 2 | 0.1% |
| | ANNALS OF INTENSIVE CARE | 1,930 | 8 | 0.2% |
| | BIOLOGY OF SEX DIFFERENCES | 2,285 | 2 | 0.1% |
| | BMC ANESTHESIOLOGY | 2,215 | 5 | 0.1% |
| | BMC CANCER | 2,215 | 136 | 3.9% |
| | BMC CARDIOVASCULAR DISORDERS | 2,215 | 21 | 0.6% |
| | BMC COMPLEMENTARY AND ALTERNATIVE MEDICINE | 2,215 | 23 | 0.7% |
| | BMC ENDOCRINE DISORDERS | 2,215 | 11 | 0.3% |
| | BMC FAMILY PRACTICE | 2,215 | 12 | 0.3% |
| | BMC GASTROENTEROLOGY | 2,215 | 23 | 0.7% |
| | BMC GERIATRICS | 2,215 | 17 | 0.5% |
| | BMC HEALTH SERVICES RESEARCH | 2,215 | 116 | 3.4% |
| | BMC MEDICAL IMAGING | 2,215 | 3 | 0.1% |
| | BMC MEDICAL INFORMATICS AND DECISION MAKING | 2,215 | 53 | 1.5% |
| | BMC MEDICAL RESEARCH METHODOLOGY | 2,215 | 29 | 0.8% |
| | BMC MEDICINE | 2,650 | 54 | 1.6% |
| | BMC MUSCULOSKELETAL DISORDERS | 2,215 | 49 | 1.4% |
| | BMC NEPHROLOGY | 2,215 | 31 | 0.9% |
| | BMC OPHTHALMOLOGY | 2,215 | 9 | 0.3% |

*

**Table 4** (*continued*)

| Discipline | Journal | APC | Number | Percent |
|---|---|---|---|---|
| | BMC ORAL HEALTH | 2,215 | 6 | 0.2% |
| | BMC PEDIATRICS | 2,215 | 36 | 1.0% |
| | BMC PREGNANCY AND CHILDBIRTH | 2,215 | 41 | 1.2% |
| | BMC PULMONARY MEDICINE | 2,215 | 13 | 0.4% |
| | BMC SURGERY | 2,215 | 2 | 0.1% |
| | BMC UROLOGY | 2,215 | 5 | 0.1% |
| | BMC WOMENS HEALTH | 2,215 | 10 | 0.3% |
| | CANCER MEDICINE | 2,250 | 14 | 0.4% |
| | CARDIOVASCULAR DIABETOLOGY | 2,185 | 12 | 0.3% |
| | CARDIOVASCULAR ULTRASOUND | 1,960 | 4 | 0.1% |
| | CLINICAL EPIGENETICS | 2,545 | 1 | 0.0% |
| | CLINICAL INTERVENTIONS IN AGING | 2,200 | 13 | 0.4% |
| | DIABETOLOGY & METABOLIC SYNDROME | 2,215 | 3 | 0.1% |
| | DIAGNOSTIC PATHOLOGY | 2,215 | 9 | 0.3% |
| | DISEASE MARKERS | 1,500 | 13 | 0.4% |
| | EVIDENCE-BASED COMPLEMENTARY AND ALTERNATIVE MEDICINE | 2,000 | 68 | 2.0% |
| | FRONTIERS IN PHARMACOLOGY | 2,194 | 24 | 0.7% |
| | GASTROENTEROLOGY RESEARCH AND PRACTICE | 1,500 | 20 | 0.6% |
| | GUT PATHOGENS | 2,250 | 4 | 0.1% |
| | HEAD & FACE MEDICINE | 2,215 | 1 | 0.0% |
| | HEALTH AND QUALITY OF LIFE OUTCOMES | 2,215 | 33 | 1.0% |
| | INFECTIOUS AGENTS AND CANCER | 1,960 | 2 | 0.1% |
| | INTERNATIONAL JOURNAL OF CHRONIC OBSTRUCTIVE PULMONARY DISEASE | 1,865 | 12 | 0.3% |
| | INTERNATIONAL JOURNAL OF ENDOCRINOLOGY | 1,500 | 11 | 0.3% |
| | ITALIAN JOURNAL OF PEDIATRICS | 1,960 | 2 | 0.1% |
| | JOURNAL OF CARDIOTHORACIC SURGERY | 2,250 | 25 | 0.7% |
| | JOURNAL OF CARDIOVASCULAR MAGNETIC RESONANCE | 1,960 | 41 | 1.2% |
| | JOURNAL OF DIABETES RESEARCH | 1,500 | 3 | 0.1% |
| | JOURNAL OF ETHNOBIOLOGY AND ETHNOMEDICINE | 1,960 | 3 | 0.1% |
| | JOURNAL OF EXPERIMENTAL & CLINICAL CANCER RESEARCH | 2,075 | 11 | 0.3% |
| | JOURNAL OF FOOT AND ANKLE RESEARCH | 1,960 | 2 | 0.1% |
| | JOURNAL OF HEMATOLOGY & ONCOLOGY | 2,250 | 21 | 0.6% |
| | JOURNAL OF OPHTHALMOLOGY | 1,500 | 14 | 0.4% |
| | JOURNAL OF ORTHOPAEDIC SURGERY AND RESEARCH | 2,545 | 9 | 0.3% |
| | JOURNAL OF OTOLARYNGOLOGY-HEAD & NECK SURGERY | 1,960 | 30 | 0.9% |
| | JOURNAL OF TRANSLATIONAL MEDICINE | 2,215 | 82 | 2.4% |

**Table 4** (*continued*)

| Discipline | Journal | APC | Number | Percent |
|---|---|---|---|---|
| | MALARIA JOURNAL | 2,140 | 124 | 3.6% |
| | MARINE DRUGS | 2,023 | 12 | 0.3% |
| | MULTIDISCIPLINARY RESPIRATORY MEDICINE | 1,960 | 1 | 0.0% |
| | NUTRITION & DIABETES | 3,300 | 14 | 0.4% |
| | ONCOTARGETS AND THERAPY | 2,200 | 5 | 0.1% |
| | ORPHANET JOURNAL OF RARE DISEASES | 2,450 | 24 | 0.7% |
| | PAKISTAN JOURNAL OF MEDICAL SCIENCES | 71 | 2 | 0.1% |
| | PARTICLE AND FIBRE TOXICOLOGY | 1,960 | 15 | 0.4% |
| | PATIENT PREFERENCE AND ADHERENCE | 2,200 | 28 | 0.8% |
| | PEDIATRIC RHEUMATOLOGY | 1,960 | 16 | 0.5% |
| | PLOS MEDICINE | 2,900 | 64 | 1.9% |
| | PLOS ONE | 1,350 | 1,725 | 49.9% |
| | RADIATION ONCOLOGY | 1,960 | 26 | 0.8% |
| | REPRODUCTIVE HEALTH | 2,215 | 5 | 0.1% |
| | RESPIRATORY RESEARCH | 2,625 | 27 | 0.8% |
| | SCANDINAVIAN JOURNAL OF TRAUMA RESUSCITATION & EMERGENCY MEDICI | 2,150 | 2 | 0.1% |
| | SCIENTIFIC REPORTS | 1,350 | 31 | 0.9% |
| | SCIENTIFIC WORLD JOURNAL | 1,200 | 12 | 0.3% |
| | THERANOSTICS | 1,168 | 15 | 0.4% |
| | THERAPEUTICS AND CLINICAL RISK MANAGEMENT | 2,200 | 6 | 0.2% |
| | THESCIENTIFICWORLDJOURNAL | 1,200 | 6 | 0.2% |
| | TOXINS | 1,124 | 7 | 0.2% |
| | TRIALS | 1,960 | 53 | 1.5% |
| | WORLD JOURNAL OF EMERGENCY SURGERY | 1,960 | 11 | 0.3% |
| | WORLD JOURNAL OF SURGICAL ONCOLOGY | 2,250 | 21 | 0.6% |
| Biomedical Research Disciplines | AIDS RESEARCH AND THERAPY | 2,165 | 23 | 0.4% |
| | ALGORITHMS FOR MOLECULAR BIOLOGY | 1,960 | 12 | 0.2% |
| | ALLERGY ASTHMA AND CLINICAL IMMUNOLOGY | 1,960 | 9 | 0.2% |
| | BEHAVIORAL AND BRAIN FUNCTIONS | 2,215 | 13 | 0.2% |
| | BIOMEDICAL ENGINEERING ONLINE | 2,215 | 17 | 0.3% |
| | BMC CELL BIOLOGY | 2,215 | 15 | 0.3% |
| | BMC DEVELOPMENTAL BIOLOGY | 2,215 | 23 | 0.4% |
| | BMC GENETICS | 2,215 | 31 | 0.6% |
| | BMC GENOMICS | 2,215 | 354 | 6.4% |
| | BMC IMMUNOLOGY | 2,215 | 19 | 0.3% |
| | BMC INFECTIOUS DISEASES | 2,215 | 66 | 1.2% |
| | BMC MEDICAL GENETICS | 2,215 | 60 | 1.1% |
| | BMC MEDICAL GENOMICS | 2,215 | 47 | 0.9% |
| | BMC MICROBIOLOGY | 2,215 | 60 | 1.1% |
| | BMC MOLECULAR BIOLOGY | 2,215 | 13 | 0.2% |
| | BMC NEUROLOGY | 2,215 | 42 | 0.8% |
| | BMC NEUROSCIENCE | 2,215 | 46 | 0.8% |

| Discipline | Journal | APC | Number | Percent |
|---|---|---|---|---|
| | BRAIN AND BEHAVIOR | 2,500 | 18 | 0.3% |
| | CANCER CELL INTERNATIONAL | 2,125 | 9 | 0.2% |
| | CELL COMMUNICATION AND SIGNALING | 2,500 | 7 | 0.1% |
| | CELL DIVISION | 2,125 | 8 | 0.1% |
| | COMPUTATIONAL INTELLIGENCE AND NEUROSCIENCE | 1,000 | 4 | 0.1% |
| | EPIGENETICS & CHROMATIN | 2,545 | 16 | 0.3% |
| | EVODEVO | 2,545 | 6 | 0.1% |
| | FRONTIERS IN AGING NEUROSCIENCE | 2,194 | 15 | 0.3% |
| | FRONTIERS IN BEHAVIORAL NEUROSCIENCE | 2,194 | 28 | 0.5% |
| | FRONTIERS IN CELLULAR NEUROSCIENCE | 2,194 | 21 | 0.4% |
| | FRONTIERS IN COMPUTATIONAL NEUROSCIENCE | 2,194 | 30 | 0.5% |
| | FRONTIERS IN HUMAN NEUROSCIENCE | 2,194 | 154 | 2.8% |
| | FRONTIERS IN MICROBIOLOGY | 2,194 | 94 | 1.7% |
| | FRONTIERS IN MOLECULAR NEUROSCIENCE | 2,194 | 19 | 0.3% |
| | FRONTIERS IN NEURAL CIRCUITS | 2,194 | 38 | 0.7% |
| | FRONTIERS IN NEUROANATOMY | 2,194 | 16 | 0.3% |
| | FRONTIERS IN NEUROINFORMATICS | 2,194 | 16 | 0.3% |
| | G3-GENES GENOMES GENETICS | 1,950 | 68 | 1.2% |
| | GENES | 562 | 8 | 0.1% |
| | IMMUNITY & AGEING | 1,960 | 4 | 0.1% |
| | JOURNAL OF CELLULAR AND MOLECULAR MEDICINE | 2,500 | 8 | 0.1% |
| | JOURNAL OF INFLAMMATION-LONDON | 2,085 | 10 | 0.2% |
| | JOURNAL OF NEUROENGINEERING AND REHABILITATION | 2,215 | 29 | 0.5% |
| | JOURNAL OF NEUROINFLAMMATION | 2,285 | 46 | 0.8% |
| | MBIO | 3,000 | 121 | 2.2% |
| | MEDIATORS OF INFLAMMATION | 1,500 | 21 | 0.4% |
| | MOBILE DNA | 2,545 | 6 | 0.1% |
| | MOLECULAR AUTISM | 2,545 | 18 | 0.3% |
| | MOLECULAR BRAIN | 2,000 | 15 | 0.3% |
| | MOLECULAR CANCER | 2,215 | 54 | 1.0% |
| | MOLECULAR CYTOGENETICS | 1,960 | 8 | 0.1% |
| | MOLECULAR NEURODEGENERATION | 2,420 | 47 | 0.9% |
| | MOLECULAR PAIN | 2,625 | 26 | 0.5% |
| | MOLECULAR SYSTEMS BIOLOGY | 4,114 | 85 | 1.5% |
| | NEURAL DEVELOPMENT | 2,545 | 31 | 0.6% |
| | NEUROPSYCHIATRIC DISEASE AND TREATMENT | 2,200 | 25 | 0.5% |
| | NEUROSIGNALS | 1,798 | 4 | 0.1% |
| | OXIDATIVE MEDICINE AND CELLULAR LONGEVITY | 1,500 | 10 | 0.2% |
| | PARASITES & VECTORS | 2,015 | 28 | 0.5% |

| Discipline | Journal | APC | Number | Percent |
|---|---|---|---|---|
| | PLOS GENETICS | 2,250 | 322 | 5.8% |
| | PLOS NEGLECTED TROPICAL DISEASES | 2,250 | 64 | 1.2% |
| | PLOS ONE | 1,350 | 2,639 | 47.9% |
| | PLOS PATHOGENS | 2,250 | 242 | 4.4% |
| | RETROVIROLOGY | 2,215 | 65 | 1.2% |
| | SCIENTIFIC REPORTS | 1,350 | 53 | 1.0% |
| | SCIENTIFIC WORLD JOURNAL | 1,200 | 6 | 0.1% |
| | THESCIENTIFICWORLDJOURNAL | 1,200 | 8 | 0.1% |
| | VIROLOGY JOURNAL | 2,215 | 55 | 1.0% |
| | VIRUSES-BASEL | 1,573 | 36 | 0.7% |
| Life Sciences | BIODATA MINING | 1,960 | 1 | 0.0% |
| | BIOLOGICAL PROCEDURES ONLINE | 2,250 | 2 | 0.1% |
| | BIOLOGY DIRECT | 2,215 | 12 | 0.5% |
| | BIOMED RESEARCH INTERNATIONAL | 1,500 | 38 | 1.7% |
| | BIOTECHNOLOGY FOR BIOFUELS | 2,545 | 22 | 1.0% |
| | BMC BIOCHEMISTRY | 2,215 | 13 | 0.6% |
| | BMC BIOLOGY | 2,650 | 36 | 1.6% |
| | BMC BIOPHYSICS | 2,215 | 4 | 0.2% |
| | BMC BIOTECHNOLOGY | 2,215 | 20 | 0.9% |
| | BMC EVOLUTIONARY BIOLOGY | 2,215 | 124 | 5.4% |
| | BMC PLANT BIOLOGY | 2,215 | 61 | 2.7% |
| | BMC STRUCTURAL BIOLOGY | 2,215 | 8 | 0.3% |
| | BMC SYSTEMS BIOLOGY | 2,215 | 100 | 4.4% |
| | BMC VETERINARY RESEARCH | 2,215 | 13 | 0.6% |
| | CELL AND BIOSCIENCE | 2,015 | 8 | 0.3% |
| | ELECTRONIC JOURNAL OF BIOTECHNOLOGY | 1,100 | 2 | 0.1% |
| | EUROPEAN JOURNAL OF HISTOCHEMISTRY | 1,028 | 2 | 0.1% |
| | FOOD & NUTRITION RESEARCH | 1,645 | 1 | 0.0% |
| | FORESTS | 899 | 13 | 0.6% |
| | FRONTIERS IN PHYSIOLOGY | 2,194 | 71 | 3.1% |
| | FRONTIERS IN PLANT SCIENCE | 2,194 | 59 | 2.6% |
| | FRONTIERS IN ZOOLOGY | 2,385 | 7 | 0.3% |
| | GENETICS SELECTION EVOLUTION | 1,755 | 3 | 0.1% |
| | INTERNATIONAL JOURNAL OF BEHAVIORAL NUTRITION AND PHYSICAL ACTI | 2,500 | 38 | 1.7% |
| | JOURNAL OF BIOLOGICAL ENGINEERING | 2,040 | 11 | 0.5% |
| | JOURNAL OF BIOMEDICAL SEMANTICS | 1,960 | 8 | 0.3% |
| | JOURNAL OF NANOBIOTECHNOLOGY | 2,215 | 1 | 0.0% |
| | JOURNAL OF OVARIAN RESEARCH | 1,960 | 6 | 0.3% |
| | JOURNAL OF PHYSIOLOGICAL ANTHROPOLOGY | 1,170 | 1 | 0.0% |
| | JOURNAL OF RADIATION RESEARCH | 1,371 | 6 | 0.3% |
| | JOURNAL OF THE INTERNATIONAL SOCIETY OF SPORTS NUTRITION | 2,215 | 6 | 0.3% |
| | LIPIDS IN HEALTH AND DISEASE | 2,215 | 19 | 0.8% |

| Discipline | Journal | APC | Number | Percent |
|---|---|---|---|---|
| | MICROBIAL CELL FACTORIES | 1,960 | 19 | 0.8% |
| | NUTRIENTS | 1,349 | 27 | 1.2% |
| | NUTRITION & METABOLISM | 2,060 | 16 | 0.7% |
| | NUTRITION JOURNAL | 2,385 | 36 | 1.6% |
| | ONCOGENESIS | 3,300 | 5 | 0.2% |
| | PLANT METHODS | 1,990 | 12 | 0.5% |
| | PLOS BIOLOGY | 2,900 | 118 | 5.2% |
| | PLOS COMPUTATIONAL BIOLOGY | 2,250 | 194 | 8.5% |
| | PLOS ONE | 1,350 | 1,004 | 43.9% |
| | PROTEOME SCIENCE | 2,215 | 12 | 0.5% |
| | REDOX BIOLOGY | 1,500 | 4 | 0.2% |
| | REPRODUCTIVE BIOLOGY AND ENDOCRINOLOGY | 2,060 | 25 | 1.1% |
| | SCIENTIFIC REPORTS | 1,350 | 29 | 1.3% |
| | SCIENTIFIC WORLD JOURNAL | 1,200 | 2 | 0.1% |
| | SYMMETRY-BASEL | 562 | 1 | 0.0% |
| | VETERINARY RESEARCH | 1,755 | 12 | 0.5% |
| | ZOOKEYS | 411 | 54 | 2.4% |
| **Chemistry** | INTERNATIONAL JOURNAL OF MOLECULAR SCIENCES | 1,798 | 47 | 24.9% |
| | INTERNATIONAL JOURNAL OF POLYMER SCIENCE | 1,200 | 2 | 1.1% |
| | MOLECULES | 2,023 | 28 | 14.8% |
| | PLOS ONE | 1,350 | 44 | 23.3% |
| | POLYMERS | 1,349 | 6 | 3.2% |
| | SCIENTIFIC REPORTS | 1,350 | 17 | 9.0% |
| | SENSORS | 2,023 | 44 | 23.3% |
| | SYMMETRY-BASEL | 562 | 1 | 0.5% |
| **Physics and Astronomy** | ADVANCES IN ASTRONOMY | 1,000 | 7 | 5.0% |
| | ADVANCES IN CONDENSED MATTER PHYSICS | 1,200 | 1 | 0.7% |
| | ADVANCES IN MATHEMATICAL PHYSICS | 1,200 | 2 | 1.4% |
| | ENTROPY | 1,349 | 17 | 12.2% |
| | INTERNATIONAL JOURNAL OF PHOTOENERGY | 1,200 | 5 | 3.6% |
| | NANOSCALE RESEARCH LETTERS | 1,385 | 17 | 12.2% |
| | PLOS ONE | 1,350 | 34 | 24.5% |
| | SCIENTIFIC REPORTS | 1,350 | 56 | 40.3% |
| **Engineering** | ADVANCES IN ELECTRICAL AND COMPUTER ENGINEERING | 274 | 1 | 0.2% |
| | ADVANCES IN MATERIALS SCIENCE AND ENGINEERING | 1,200 | 3 | 0.7% |
| | ADVANCES IN MECHANICAL ENGINEERING | 1,500 | 5 | 1.1% |
| | APPLIED SCIENCES-BASEL | 562 | 1 | 0.2% |
| | BMC BIOINFORMATICS | 2,215 | 259 | 59.4% |
| | CRYSTALS | 562 | 3 | 0.7% |
| | ENERGIES | 1,349 | 11 | 2.5% |

**Table 4** (*continued*)

| Discipline | Journal | APC | Number | Percent |
|---|---|---|---|---|
| | EURASIP JOURNAL ON ADVANCES IN SIGNAL PROCESSING | 1,455 | 8 | 1.8% |
| | EURASIP JOURNAL ON IMAGE AND VIDEO PROCESSING | 1,145 | 2 | 0.5% |
| | EVOLUTIONARY BIOINFORMATICS | 1,980 | 6 | 1.4% |
| | FRONTIERS IN NEUROROBOTICS | 2,194 | 1 | 0.2% |
| | INTERNATIONAL JOURNAL OF ANTENNAS AND PROPAGATION | 1,500 | 2 | 0.5% |
| | INTERNATIONAL JOURNAL OF DISTRIBUTED SENSOR NETWORKS | 1,500 | 4 | 0.9% |
| | JOURNAL OF NANOMATERIALS | 1,200 | 12 | 2.8% |
| | JOURNAL OF SENSORS | 1,000 | 1 | 0.2% |
| | MATERIALS | 1,573 | 21 | 4.8% |
| | MATHEMATICAL PROBLEMS IN ENGINEERING | 1,200 | 10 | 2.3% |
| | METALS | 337 | 1 | 0.2% |
| | MICROMACHINES | 562 | 3 | 0.7% |
| | NANOMATERIALS | 337 | 1 | 0.2% |
| | OPTICAL MATERIALS EXPRESS | 1,350 | 11 | 2.5% |
| | PLOS ONE | 1,350 | 38 | 8.7% |
| | SCIENTIFIC REPORTS | 1,350 | 12 | 2.8% |
| | THEORETICAL BIOLOGY AND MEDICAL MODELLING | 2,215 | 20 | 4.6% |
| **Earth Science** | ADVANCES IN METEOROLOGY | 1,200 | 6 | 0.9% |
| | ATMOSPHERE | 562 | 4 | 0.6% |
| | BMC ECOLOGY | 2,215 | 3 | 0.5% |
| | ECOLOGY AND EVOLUTION | 1,950 | 44 | 6.6% |
| | ENVIRONMENTAL HEALTH | 2,040 | 92 | 13.9% |
| | ENVIRONMENTAL RESEARCH LETTERS | 1,920 | 83 | 12.5% |
| | INTERNATIONAL JOURNAL OF ENVIRONMENTAL RESEARCH AND PUBLIC HEAL | 1,798 | 88 | 13.3% |
| | MINERALS | 337 | 4 | 0.6% |
| | PLOS ONE | 1,350 | 285 | 42.9% |
| | REMOTE SENSING | 1,349 | 20 | 3.0% |
| | SCIENTIFIC REPORTS | 1,350 | 11 | 1.7% |
| | SCIENTIFIC WORLD JOURNAL | 1,200 | 2 | 0.3% |
| | SUSTAINABILITY | 1,124 | 14 | 2.1% |
| | WATER | 1,124 | 8 | 1.2% |
| **Business and Economics** | PLOS ONE | 1,350 | 11 | 100.0% |
| **Psychiatry/Psychology** | ANNALS OF GENERAL PSYCHIATRY | 2,545 | 7 | 1.9% |
| | BMC PSYCHIATRY | 2,215 | 36 | 9.7% |
| | EUROPEAN JOURNAL OF PSYCHOTRAUMATOLOGY | 1,303 | 6 | 1.6% |
| | FRONTIERS IN PSYCHOLOGY | 2,194 | 145 | 38.9% |
| | INTERNATIONAL JOURNAL OF MENTAL HEALTH SYSTEMS | 1,960 | 3 | 0.8% |
| | PLOS ONE | 1,350 | 171 | 45.8% |

**Table 4** (*continued*)

| Discipline | Journal | APC | Number | Percent |
|---|---|---|---|---|
| | SCIENTIFIC REPORTS | 1,350 | 3 | 0.8% |
| | THESCIENTIFICWORLDJOURNAL | 1,200 | 2 | 0.5% |
| **Social Science** | BMC INTERNATIONAL HEALTH AND HUMAN RIGHTS | 2,215 | 14 | 1.9% |
| | BMC MEDICAL EDUCATION | 2,215 | 30 | 4.1% |
| | BMC MEDICAL ETHICS | 2,215 | 5 | 0.7% |
| | BMC PALLIATIVE CARE | 2,215 | 5 | 0.7% |
| | BMC PUBLIC HEALTH | 2,215 | 235 | 32.4% |
| | GLOBALIZATION AND HEALTH | 2,215 | 29 | 4.0% |
| | HARM REDUCTION JOURNAL | 2,545 | 42 | 5.8% |
| | HEALTH RESEARCH POLICY AND SYSTEMS | 1,960 | 6 | 0.8% |
| | HUMAN RESOURCES FOR HEALTH | 2,545 | 16 | 2.2% |
| | IMPLEMENTATION SCIENCE | 2,300 | 52 | 7.2% |
| | INTERNATIONAL JOURNAL FOR EQUITY IN HEALTH | 2,040 | 22 | 3.0% |
| | INTERNATIONAL JOURNAL OF CIRCUMPOLAR HEALTH | 686 | 16 | 2.2% |
| | INTERNATIONAL JOURNAL OF HEALTH GEOGRAPHICS | 1,960 | 21 | 2.9% |
| | INTERNATIONAL JOURNAL OF QUALITATIVE STUDIES ON HEALTH AND WELL | 1,234 | 1 | 0.1% |
| | JOURNAL OF OCCUPATIONAL MEDICINE AND TOXICOLOGY | 2,085 | 3 | 0.4% |
| | MEDICAL EDUCATION ONLINE | 1,166 | 10 | 1.4% |
| | PLOS ONE | 1,350 | 192 | 26.4% |
| | POPULATION HEALTH METRICS | 1,960 | 11 | 1.5% |
| | SCIENTIFIC REPORTS | 1,350 | 1 | 0.1% |
| | SCIENTIFIC WORLD JOURNAL | 1,200 | 1 | 0.1% |
| | SUBSTANCE ABUSE TREATMENT PREVENTION AND POLICY | 1,960 | 13 | 1.8% |
| | SYMMETRY-BASEL | 562 | 1 | 0.1% |

### Funding

This work was in part funded by the Andrew W. Mellon Foundation through a grant to The University of California, Davis. The funders had no role in study design, data collection and analysis, decision to publish, or preparation of the manuscript.

### Grant Disclosures

The following grant information was disclosed by the authors:
Andrew W. Mellon Foundation.

### Competing Interests

David Solomon is a former Academic Editor for PeerJ.

## Author Contributions

- David Solomon conceived and designed the experiments, performed the experiments, analyzed the data, wrote the paper, prepared figures and/or tables, reviewed drafts of the paper.
- Bo-Christer Björk conceived and designed the experiments, reviewed drafts of the paper.

## Data Availability

The raw data has been supplied as Data S1.

## Supplemental Information

Supplemental information for this article can be found online at http://dx.doi.org/10.7717/peerj.2264#supplemental-information.

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
