# Peer review of "(untitled)"

_PeerJ, doi:10.7717/peerj.2264_

## Round 0.1 · original submission · Minor Revisions

· Academic Editor

Minor Revisions

I think the reviewers have articulated their minor issues quite clearly. I would tend to reiterate the sentiment that tell-tale signs of market dysfunction such as hyperinflation in APCs deserve a special mention.

·

Basic reporting

TWO MAJOR ISSUES
Clarification request re apparent conflict of interest

The description of the Pay It Forward project in the second paragraph of the Introduction is useful. However, the role of the authors in this project needs to be made very clear. The Pay It Forward website states that “also working with experts in Scholarly Communications… Professor David Solomon (Michigan State University), Professor Bo-Christer Björk “. At minimum, the authors’ involvement in the project must be explained. It is particularly important, if either of the authors have worked as a consultant or is involved in consultancy work in this area this should be stated at the very beginning of the introduction. Otherwise, consider how this kind of arrangement would be perceived if an academic author receiving funding for research from a pharmaceutical company and failing to disclose this is their publication would look.

It appears to me that the authors are echoing the perspective of the Pay it Forward members. That’s fine, but should be made clear. One should say things like, “we work for the Pay it Forward project members (or are part of the Pay It Forward Team) who have decided that APC is the way to go for OA publishing. We concur because…”

Important omission – APC prices much higher than previous reports and possibility of price rises far beyond inflation.

The average APC of $1,865 USD for fully OA journals found in this study is significantly higher than the average global APCs reported by the authors and by Morrison et al. of under $1,000, and also significantly higher than the authors’ studies of 2012 and 2014 as reported at the bottom of p. 3: “The average APC in this subset of journals was 1,292 USD in the fall of 2012 and had increased to 1,418 USD by the fall of 2014”. An increase from an average of $1,292 to $1,865 in 4 years is a 44% price increase during years when the inflation rate is hovering around 1 – 2% and has even been negative some years in the EU. From $1,418 in 2014 to $1,865 in 2016 is an increase of over 31% in 2 years. This data would tend to suggest that the same market dysfunction that was observed in subscription pricing in recent decades might simply continue into an APC flip. It is important for people considering support for open access to be aware of this; this should be highlighted and discussed in the article.

Experimental design

No comments

Validity of the findings

p. 2
Abstract: Open access (OA) publishing via article processing charges (APCs) is growing as an alternative to subscription publishing
Comment: it is important to distinguish between open access publishing and APCs. This is just one model of funding that is used by a minority of fully open access journals. For example, our team recently found that De Gruyter, a commercial publisher that has in recent years become one of the 3 largest OA journal publishers as measured by journal numbers, uses APCs for only 2% of its journals. The rest are all sponsored.

Conclusions: Full OA journal APCs average a little under 2,000 USD while hybrid articles average about 3,000 USD. There is a lack of information on discipline differences in APCs due to the concentration of APC funded publications in a few fields and the multidisciplinary nature of research.
Comments:
1. It is important to emphasize that these averages are for the articles covered in this study. Note that this figure is approximately double the global average APC amount.
2. A concentration of APC funded publications in a few fields is a disciplinary difference.

Introduction

Results p. 7
Re point 2: There do not appear to be large discipline differences in APCs. This likely reflects the limitations of the data available.
Comment: it is good to point out the limitation, however I don’t think your data support even this softened conclusion. Perhaps change to: “due to concentration of APC payments in particular fields and the multidisciplinary nature of the articles, we were not able to observe any large discipline differences in APCs”.
Proofreading is needed.

Comments for the author

Article Processing Charges for Open Access Publication – The Situation in 2016
Review by: Heather Morrison

This is an important study which I enjoyed reviewing - this merits publication with some minor revisions (in terms of work required) that are nonetheless extremely important.

·

Basic reporting

No Comments

Experimental design

The investigation is methodically sound, clearly described, and replicable. It builds upon former analyses and studies of such kind in a comprehensive manner reflecting the state of the art.
However, the raised issue of exploring the feasibility of transitioning from paying subscriptions to funding APCs at research intense institutions (referring to the PIF project) weren’t met by the investigation, which is a bit disappointing. For example, information on the annual publishing output and its distribution over disciplines and publishers of the PIF partner institutions is missing. On that basis, cost modelling at given APC levels could have been presented.
Moreover, to address the transition aspect, a connection to the subscription spending of the institutions would be essential. The actual approach to investigate on the the evolvement of the current APC spending in order to gain a deeper understanding of the costs of the oa transition seems questionable. What we rather do know at the moment, however, are the expenditures on licenses and subscriptions. These costs could probably provide a better basis for such predictions.
I would therefore like to suggest addressing the research question more clearly (state, that it is more or less a review of former similar analyses) and/ or refer to the necessity of including subscription spending in further investigations.

Validity of the findings

No Comments

---

## Round 0.2 · Minor Revisions

· Academic Editor

Minor Revisions

One reviewer had some suggestions which I think are worth incorporating into the manuscript. With regard to potential conflicts of interests, they are in themselves not a major issue, as long as they are fully disclosed. This information may be helpful:

Responsible Research Publication, International standards for authors, from Second World Conference on Research Integrity, available for download from here: http://publicationethics.org/resources/international-standards-for-editors-and-authors

See Section 5 - transparency. There is nothing wrong with working as a consultant and using data (assuming permission) to write an academic article. It is important for the reader to know that this is the case.

·

Basic reporting

Review round 2 – 3 points

1. Re: Comment: Clarification request re apparent conflict of interest
It appears that you have provided some information to PeerJ in the declarations that I have not seen.

Suggestion: add this statement “This work was in part funded by the Andrew W. Mellon Foundation through a grant to The University of California, Davis” at the end of the second paragraph. Because U C Davis is an active partner in PIF, this is important context for the reader. A repeat of this in an acknowledgement section would be in order.

2. Suggestion: re-read / re-consider your introductory paragraph.

The first paragraph of the introduction is weak – it appears to start with a conclusion and prescription and without sufficient background. Two suggestions to make this stronger – I recommend the first approach:

2A. Make the activist background clear. This is not neutral research; we are trying to actively transition scholarship to open access. A couple of suggestions for references to draw from:

This statement from the U of C Davis’ grant application: “Our study focuses on APC-funded scholarship because we observe that should this model become the predominant open access funding model for the majority of journals in which authors at large North American research institution like the University of California publish, there is a significant risk that we would be unable to continue to sustain scholarly communication” (see icis.ucdavis.edu/wp-content/uploads/.../UC-Pay-It-Forward-narrative-2014-FINAL.pdf).

The Max Planck white paper for the Berlin 12 conference would also be a useful reference: http://pubman.mpdl.mpg.de/pubman/faces/viewItemOverviewPage.jsp?itemId=escidoc:2148961

2B. Provide data to back-up your claims in the original introductory paragraph, i.e.:

_ data to support the claim that “APCs have become the dominant business model for professional Open Access (OA) publishing” (define professional OA publishing and provide figures to back-up your statement that APCs are dominant)
- data to support the claim that “many other viable business models for funding OA journals and the number of articles published without author fees also continues to grow abet at a slower rate”, i.e. please provide data on the rate of increase of number of articles published with and without author fees

3. Re: Important omission – APC prices much higher than previous reports and possibility of price rises far beyond inflation.
Comment: your rebuttal reacts to the global estimated APCs and misses the point about your own research on a subset of journals that appears to be similar to what is studied here of 1,292 USD in fall of 2012 increasing to 1,418 USD. If the average APCs for the most similar types of journals has increased from $1,292 USD in 2014 to $2,000 in 2016, that is a huge inflationary increase. This article is likely to be read by people who need to make decisions about whether or not to support APCs. This is important information. If people decide to support APCs and budget according to the average you report and there is this kind of inflation going on, they will under-budget and likely run out of funds part-way through the year. It is possible that there is something else going on here, but I think your article would be much stronger if you were to address this.

Experimental design

see general notes

Validity of the findings

see general notes.

Comments for the author

Addressing these 3 points doesn't have to be a lot of work; regarding the third (potential indicator of high inflationary increases), a note highlighting this and indicating it is something to look into would suffice. I know how annoying this is, but this work would make the article stronger.

·

Basic reporting

no comments

Experimental design

no comments

Validity of the findings

no comments

Comments for the author

no comments

---

## Round 0.3 · accepted · Accept

· Academic Editor

Accept

Thank you very much for your submission and congratulations on this important contribution.

·

Basic reporting

This is important and original work which exceeds PeerJ's requirements. The authors have fully addressed all required and suggested changes from the first two rounds of review. Recommendation: accept as is. I won't fill out all of the sections as I have nothing to add at this point.

Experimental design

Accept as is.

Validity of the findings

No limitations to validity aside from those inherent and unavoidable due to research design.

Comments for the author

Kudos - another valuable contribution to move us forward in this area, look forward to your next projects!